# Evaluation of the Cytotoxic Activity of the *Usnea barbata* (L.) F. H. Wigg Dry Extract

**DOI:** 10.3390/molecules25081865

**Published:** 2020-04-17

**Authors:** Violeta Popovici, Laura Adriana Bucur, Verginica Schröder, Daniela Gherghel, Cosmin Teodor Mihai, Aureliana Caraiane, Florin Ciprian Badea, Gabriela Vochița, Victoria Badea

**Affiliations:** 1Departament of Microbiology and Immunology, Faculty of Dental Medicine, Ovidius University of Constanta, 7 Ilarie Voronca Street, 900684 Constanta, Romania; popovicivioleta@gmail.com (V.P.); badea_victoria@yahoo.com (V.B.); 2Department of Pharmacognosy, Faculty of Pharmacy, Ovidius University of Constanta, 6 Capitan Al. Serbanescu Street, 900001 Constanta, Romania; adrianabucur@yahoo.com; 3Department of Cellular and Molecular Biology, Faculty of Pharmacy, Ovidius University of Constanta, 6 Capitan Al. Serbanescu Street, 900001 Constanta, Romania; 4Institute of Biological Research Iasi, branch of NIRDBS, 47 Lascar Catargi Street, 700107 Iasi, Romania; mihai.cosmin.teo@gmail.com (C.T.M.); gabrielacapraru@yahoo.com (G.V.); 5Advanced Centre for Research and Development in Experimental Medicine (CEMEX), “Grigore T. Popa” University of Medicine and Pharmacy Iasi, 9-13 Mihail Kogalniceanu Street, 700259 Iasi, Romania; 6Department of Oral Rehabilitation, Faculty of Dental Medicine, Ovidius University of Constanta, 7 Ilarie Voronca Str., 900684 Constanta, Romania; drcaraiane@yahoo.com (A.C.); florinb.md@gmail.com (F.C.B.)

**Keywords:** *Usnea barbata* (L.) F. H. Wigg, dry extract, usnic acid, cytotoxic activity, Brine shrimp, CAL 27 cells

## Abstract

The secondary metabolites of lichens have proven to be promising sources of anticancer drugs; one of the most important of these is usnic acid, which is a phenolic compound with dibenzofuran structure that is responsible for the numerous biological actions of lichens of genus *Usnea*. As a result, in this study, we related to this phenolic secondary metabolite. The aim of the present study is the evaluation of the cytotoxic activity of *Usnea barbata* (L.) F. H. Wigg dry acetone extract (UBE). In advance, the usnic acid content was determined in various extracts of *Usnea barbata* (L.) F. H. Wigg: the liquid extracts were found in water, ethanol, acetone, and the dry acetone extract; the highest usnic acid quantity was found in the dry acetone extract. First, the cytotoxic action of UBE was assessed using Brine Shrimp Lethality (BSL) test; a significant lethal effect was obtained after 24 h of treatment at high used concentrations of UBE, and it was quantified by the high mortality rate of the *Artemia salina* (L.) larvae. Secondly, in vitro cytotoxicity of UBE was evaluated on human tongue squamous cells carcinoma, using CAL 27 (ATCC^®^ CRL-2095™) cell line. The most intense cytotoxic effect of UBE on CAL 27 cells was registered after 24 h; this response is directly proportional with the tested UBE concentrations. The obtained results have been reported regarding usnic acid content of UBE, and the data show that CAL 27 cells death was induced by apoptosis and high oxidative stress.

## 1. Introduction

The fight against cancer involves considerable scientific and financial efforts to discover new solutions for the early detection of the carcinogenic process and for the development of new specific therapeutic tools for the total destruction of cancer cells with limited effects on normal cells [1]. It is known that carcinogenesis is a complex process, involving cellular and molecular alterations, mediated by endogenous and/or exogenous factors: oxidative DNA damage, chromosomal abnormalities, and oncogene activation are responsible for the development of cancer and may be induced by prolonged oxidative stress [2]. Thus, in cancer pathology, the reactive oxygen species (ROS) have dual role, confirmed by many studies: first, ROS are cytotoxic, having an important contribution in the etiology and progression of cancer [3]. At the same time, many antitumor agents can destroy the cancer cells through intense oxidative stress, generating ROS production in high quantities [4]. Increasing ROS levels through redox modulation could in the future be an effective strategy for the selective destruction of cancer cells (not normal cells) [5]; this method is called “oxidative therapy” [6], and it was developed by inducing cytotoxic oxidation as a stress factor in cancer cells in the treatment of cancer [7].

One of the most invasive malignancies is the oral squamous cell carcinoma (OSCC), which is the most common cancer of oral cavity; it can affect any part of the oral cavity including lips, tongue, gums, buccal epithelium, and salivary glands [8]. Despite that, the oral cavity is easily accessible for the direct visual examination, and the mortality from oral cancer remains still high because in the early stages, the patients have no symptoms; being a highly invasive cancer type with a great lymphophilic character, at the first medical consultation, the patients frequently have metastatic cervical adenopathies [9]. If the tumor is discovered in the early stages and treated properly, the survival time is favorable; however, in advanced stages, the survival does not exceed two years, because the oral malignancies are extremely aggressive. The global mortality rate due to the oral cancer is estimated at about 3%–10% [10].

It is noteworthy that a significant number of new chemotherapeutics used in cancer therapy were obtained from natural sources [11].

There are still many potential plants sources of antitumor compounds, which need to be investigated, including the lichens [12]; they have a special dual structure, as a result of the symbiosis between a fungus and an algae/cyanobacteria [13]. The specific conditions in which they live determines the synthesis of numerous secondary metabolites, including most of these compounds with phenolic structure [14], which provide the lichens with optimal protection against disruptive, physical, and biological factors [15]. These lichen secondary metabolites, structurally close, exhibit many biological actions: antibacterial, antiviral, antioxidant, anti-inflammatory, cytotoxic, and antitumor [16].

*Usnea barbata* (L.) F. H. Wigg. is a fruticose lichen belonging to the *Parmeliaceae* family [17] who lives in the coniferous forests of Europe and North America, and for thousands of years, it has been used in the traditional medicine to treat various conditions [18].

The phenolic compounds [19] identified in this species were mainly depsides, depsidones, lipids, diphenylether derivatives, and dibenzofurans [20]. The most important secondary metabolite, common to all species of the genus *Usnea*, is usnic acid—C_18_H_16_O_7_-[2,6-diacetyl-7,9-dihydroxy-8,9b-dimethyldibenzofuran-1,3(2H,9bH)-dione], which is a phenolic compound with dibenzofuran structure [21] (Figure 1). Both the (+) and (−) enantiomers, as well as their racemic mixture of usnic acid, display numerous biological activities including cytotoxic activities against cancer cells [22], which are responsible for most of the biological actions of the lichens of this genus [23].

For this study, *Usnea barbata* (L.) F. H. Wigg was harvested from Calimani Mountains (Romania), at the beginning of March, from an altitude of about 900 m. In our previous studies, different extracts of *Usnea barbata* (L.) F. H. Wigg. were obtained and by High-Performance Liquid Chromatography (HPLC) [24] and Ultra-High Performance Liquid Chromatography methods (UHPLC) [25], we determined the content of the usnic acid in each extract.

Cytotoxic activity of *Usnea barbata* (L.) F.H. Wigg. dry extract was investigated on *Artemia salina* L. (Brine shrimp) larvae. After this preliminary test, the antitumor potential of UBE was evaluated in vitro on human tongue squamous cells carcinoma—CAL 27 (ATCC^®^ CRL-2095™) cell line—by assessment of cell viability and apoptotic process.

## 2. Results

### 2.1. Determination of Usnic Acid Content in Different Extracts of Usnea barbata (L.) F. H. Wigg

Usnic acid content in various extracts of *Usnea barbata* (L.) F. H. Wigg. determined by HPLC and UHPLC is shown in Table 1.

The highest content of usnic acid was found in the UBE dissolved in acetone, followed by UBE solubilized in dimethyl sulfoxide (DMSO), which was about two times lower than UBE dissolved in acetone, and the acetone macerate had eight-fold lower content of usnic acid than UBE dissolved in DMSO. The aqueous extract had the smallest content of usnic acid (0.04%) (Table 1).

### 2.2. Evaluation of UBE Cytotoxic Activity by BSL Assay

Assessment of UBE cytotoxicity by BSL assay shows correlated response with exposure time and concentration of dry extract. Thus, the passage from one larval stage to another was achieved, and consequently, the growth was not affected. Following these observations, it can be considered that mortality of *Artemia salina* L. larvae is due to the blockage of cellular activity after accumulation of UBE at this level.

Intensity of UBE cytotoxic effect on *Artemia salina* L. is correlated with the death rate of larvae in the treated samples. The results were assessed with Probit analysis, which is a method of analyzing the relationship between a stimulus and the binomial response; the Probit test shows significant lethal effects at values higher than 100 µg/mL, where LC_50_ = 164.92 µg/mL (Figure 2).

### 2.3. Effects of UBE Treatment on Morphological Characteristics of CAL 27 (ATCC^®^ CRL-2095™) Cell Line

The morphology of the CAL 27 cells has changed depending on the cell culture age, duration of the treatment, and UBE concentration used (Figure 3a–o). Thus, at initiated culture (T0), the cells have a globular appearance showing Brownian movements in the culture medium (Figure 3a). After 24 h and 48 h of cultivation, the untreated cells (Control) have normal morphological specifications in time, exhibit uniform adhesion, and grow in a monolayer shape (Figure 3b,i). The 24 h treatment with UBE (12.5, 25, 50, 100, 200, and 400 µg/mL) induced different degrees of morphological changes, amplified with the increasing concentrations of UBE, including: loss of cell adhesion, membrane shrinkage, formation of abnormal cellular wrinkle, cell fragmentation and reduction of cellular density as result of the increase in the number of dead cells and, consequently, reduction of cell viability (Figure 3c–h). These manifestations produced by the UBE exposure on CAL 27 cells become more intense after extension of the treatment up to 48 h (Figure 3j–o).

### 2.4. In Vitro Cytotoxicity of UBE on CAL 27 (ATCC^®^ CRL-2095™) Cell Line

The 24 h and 48 h treatment of CAL 27 cells was performed with 6 doses of UBE in DMSO 0.2%: 12.5, 25, 50, 100, 200, and 400 µg/mL. In vitro highlighting and confirming UBE cytotoxic effect on CAL 27 cells, using all these UBE concentrations were achieved by 3-[4,5-dimethylthiazol-2-yl]-2,5 diphenyltetrazolium bromide (MTT) assay.

#### 2.4.1. MTT Assay

The results of MTT assay are depicted in Figure 4 and Figure 5. After exposure to UBE, there are three thresholds of cell response as follows: the first is between 12.5 and 50 µg/mL doses, the second is between 100 and 200 µg/mL, and the third is at 400 µg/mL. In the first case, including DMSO 0.2%, the cell viability was over 90%, corresponding to a negligible cytotoxic effect. Beginning with 100 µg/mL UBE, there was a slight decreased cell viability, registering 62.5%, which corresponds to the induction of a cytotoxic effect of 37.5%.

The reactivity of CAL 27 cells at doses of 200 and 400 µg/mL UBE was higher, reaching an inhibition of proliferation of 45.9% and 66.6%, respectively. In order to observe the persistence in time of the antiproliferative effect, CAL 27 cells were exposed to the UBE for 48 h. Generally, there was a similar trend in sense and amplitude after 24 h treatment, and the significant inhibition of cell proliferation was registered at 200 and 400 µg/mL UBE (Figure 4).

Comparing the two exposure times, it was found that after 48 h, the cell viability slightly increased; this process may be due to the cellular self-repair mechanisms activation or, more probably, to the anarchic division of CAL 27 cells.

These obtained results are supported by morphological changes of CAL 27 cells exposed for 24 and 48 h to the same concentrations of UBE, previously evaluated and presented in Figure 3.

The cytotoxic activity of UBE was also quantified through the prism of IC 50 value, which means the concentration of the extract that inhibited the growth of cells to the level of 50% of control. The IC50 values were estimated using polynomial graphic plots of the dose–response curve for each concentration. The most intense cytotoxic effect was manifested after 24 h of treatment, where IC50 = 232.17 µg/mL (Figure 5a); after 48 h, the value of this parameter was 310.67 µg/mL (Figure 5b).

In the both contact time intervals, the non-cytotoxic effect of the solvent on CAL 27 cells was noted (Figure 4), the cell viability being 92.3% after 24 h and 92.5% after 48 h, respectively; this observation is based on the conclusion that the calculated cytotoxicity is exclusively due to UBE.

#### 2.4.2. Apoptosis Assay

The apoptosis process was evaluated after 6 and 24 h of treatment of CAL 27 cell cultures, with the two concentrations of UBE at which antitumor effects were recorded. The obtained data are represented in Figure 6. The 6 h treatment with 100 µg/mL UBE induced insignificant effects on cell viability, and correlated parameters (pre-apoptotic, apoptotic, and dead cells), were close to those of the control; at the same time, UBE 200 µg/mL reduced cell viability with a corresponding increase in the frequency of the pre-apoptotic, apoptotic, and dead cells. The 24 h treatment with both UBE concentrations caused a significant decrease of the cell viability and an important augmentation of the frequency of dead and apoptotic cells, accompanied by a minor amplitude reduction of pre-apoptotic cells. All these obtained results confirmed the expression of the cytotoxic effect through apoptosis mechanism (Figure 7A–F).

#### 2.4.3. Assessment of Antioxidant Enzyme Activity

The effect of different UBE concentrations for 24 h on superoxide dismutase (SOD) activity in CAL 27 cells, expressed in units of SOD/g protein, is illustrated in Figure 8. There is a slight stimulation of the activity of SOD by the UBE, the values recorded at 100 and 400 µg/mL being very close and about 1.25 times higher than the control.

The increase of catalase (CAT) activity may indicate a toxic accumulation of hydrogen peroxide (H_2_O_2_); SOD–CAT tandem has a great role in the antioxidant defense system. The impact of 24 h treatment with various concentrations of UBE on CAT activity, as expressed in units of CAT/g protein in CAL 27 cells led to the register of the results shown in Figure 9.

We noticed a stimulation of CAT activity, with high differences between the two UBE tested concentrations; at 100 µg/mL, this is about 1.5 times higher, while at 400 µg/mL, it is almost four times higher, compared to the control.

The interference of the 24-h treatment with UBE, in different concentrations, with the activity of peroxidase (POD) expressed in units of POD/g protein, in the CAL 27 cells, was materialized by a high enzymatic activation, almost double at the dose of 100 µg/mL and over 3 times higher at the dose of 400 µg/mL, compared to the control, as shown in Figure 10.

All the obtained experimental data are presented comparatively in Table 2.

After analysis of obtained results shown in comparative presentation in Table 2, it is observed that UBE at cytotoxic concentrations—100 µg/mL and 400 µg/mL—weakly stimulates the activity of SOD; the values corresponding to the action of the used UBE concentrations are very close. On CAT activity, the stimulation is significant, the values differ greatly—both between the two concentrations of UBE and compared with the control. The stimulation of POD activity is significant at UBE 100 µg/mL compared to the control, when a 4-fold increase in UBE concentration has in this case a smaller influence.

## 3. Discussions

For a lot of known secondary lichen metabolites, without a doubt, usnic acid is one of the most extensively studied [26]. From this perspective, the level of this dibenzofuran derivative content in lichens is very important for the medical application. Cansaran et al. [27] concluded that the maximum content of the usnic acid is found in the lichens with habitat in mountain areas, between 700 and 1500 m altitude, during winter and spring. In our study, *Usnea barbata* (L.) F. H. Wigg. was harvested from Calimani Mountains (Suceava, Romania) at the altitude of 900 m in March. In the acetone macerate, we found 2.12% usnic acid [24]; this value is very close to the one obtained by Cansaran et al. [27] in their study, which was respectively 2.16% in the same extract [27].

We noticed the dependence of the content in usnic acid from different acetone extracts of *Usnea barbata* (L.) F.H. Wigg on the extraction method [28]. The much larger percentage of usnic acid (31.59%) was measured in the dry extract dissolved in acetone, compared to the acetone macerate, in which only 2.12% of the usnic acid [24] was quantified; the results may be due to the much higher solubility of the usnic acid in acetone following the reflux.

In vitro biological experiments used the UBE dissolved in DMSO at non-toxic concentrations for living cells [29]. The treatment doses took into account the value of the usnic acid content from the dry acetonic extract dissolved in DMSO. We mentioned that when UBE was solved in DMSO, the results showed 16.53 ± 1.08% usnic acid content [25] and the presence of the other phenolic compounds (that may contribute to the cytotoxic activities of UBE) in small quantities compared to the usnic acid. This fact may suggest the idea that the usnic acid is the principal compound involved in the cytotoxic activity in the presented study—this is why we relate exclusively to this dibenzofuran derivative.

It is known that usnic acid is soluble in the both solvents, acetone and DMSO [30]; in acetone, the solubility of usnic acid is two times higher than in DMSO: 0.77 g/100 mL in acetone [31] and in DMSO it is 0.4 g/100 mL, warmed [32]. These data explain the approximately double content of usnic acid in UBE dissolved in acetone compared to that solubilized in DMSO.

The cytotoxicity assessment on *Artemia salina* L. larvae or BSL test is considered a useful tool for assessing the preliminary toxicity of the plant extracts [33]. This test is an efficient, inexpensive, and relatively fast way of detecting toxic compounds, requiring only small amounts of sample: < 20 mg [34]. This species is used in the studies for testing the cytotoxic activity of various plant extracts as well as to evaluate the toxic activity of mycotoxins [35]. In addition, BSL assay allows fast and meaningful information also in the case of teratogenic phenomena or mutagen potential [36].

The cytotoxic activity of UBE was quantified by BSL test, and this effect was directly proportional with dose and with time of action. The death rate, correlated with the moderate toxicity of UBE, was also recorded to lower concentrations, between 30 and 70 µg/mL. Significant UBE cytotoxicity was reported at concentrations of UBE higher than 100 µg/mL. This study suggests that UBE-induced mortality in *Artemia salina* L. larvae may be correlated with amplification of cytological changes (loss of cell connection, inhibition of organogenesis, and generation of cytoplasmic inclusions).

The data from the accessed literature show that values comparable to those of the present study were obtained in the other similar studies realized with other lichen species. Thus, Paudel et al. [34], in the preliminary cytotoxicity study on *Artemia salina* L., showed that for 24 lichen species in Nepal, LC_50_ was between 100 and 400 μg/mL. In another study, more recently, Ravaglia et al. [37] evaluated the toxic potential on *Artemia salina* L. larvae for six species of lichens harvested from Brazil and Antarctica; LC_50_ values between 151.0 and >600 µg/mL demonstrate that the tested extracts have a reduced toxicity on the *Artemia salina* L. larvae compared to that quantified in the present study (when LC_50_ UBE = 164.92 µg/mL).

In our study, UBE solubilized in DMSO has 16.53% usnic acid; we reported this percentage to the IC50 UBE value at 24 h (respectively 232.17 µg/mL), and we calculated that 38.38 μg/mL usnic acid is corresponding to this value.

There are many studies performed exclusively on the tumor cell lines, and in all the articles, it is emphasized that the lichen extracts have a cytotoxic action on the various tumor cells. Ranković et al. [23] tested the anticancer activity of two lichen species, *Usnea barbata* and *Toninia candida*, against FemX (human melanoma) and LS174 (human colon carcinoma) cell lines using the MTT assay. At the same time, the antitumor activity of the both its metabolites, usnic acid and norstictic acid, was evaluated, and the usnic acid was found to have the strongest anticancer action toward both cell lines with IC50 values of 12.72 and 15.66 μg/mL, which were much smaller compared to those from the present study [23]. On the apoptosis assay, to capture the peak of activity on the CAL 27 cell cultures, we used shorter exposure times (6 and 24 h, respectively) because the apoptosis of the isolated cell lines occurs within several hours [38].

The results of the present study showed that the treatment with UBE resulted in a significant reduction in the cell viability and a considerable increase in the frequency of the dead and apoptotic cells, thereby confirming the expression of the cytotoxic effect through apoptosis mechanism, depending on the concentration and the exposure time [38].

A study conducted by Rabelo et al. [39] tested the usnic acid redox properties against different ROS generated *in vitro*; they evaluated its action on the SH-SY5Y (neuroblastoma) neuronal-like cells upon hydrogen peroxide (H_2_O_2_) exposure, and they observed that the usnic acid induced cell detachment and a loss of viability of SH-SY5Y cells at higher concentrations, alone, or in the presence of H_2_O_2_. These results were related to the increase of intracellular ROS, inducing an oxidative stress scenario, which was potentiated in the presence of H_2_O_2_. The pro-oxidant properties in biological systems might be responsible for the potential neurotoxic effects of the usnic acid against neuroblastoma [39].

It is known that in the tumor cells exists a higher intrinsic oxidative stress than in the normal cells, due to their metabolic alterations; it was reported that the cancer cells have an important level of ROS, compared to the normal cells. This elevation of ROS may appear as result of an abnormal mitochondrial oxidative metabolism and can be responsible for the initiation and progression of different types of cancer [40]. However, the elevation of ROS to a great level may be lethal for tumor cells themselves [7]. In fact, several studies outlined the implication of elevated levels of H_2_O_2_ in the induction of apoptosis, and their low concentration enhanced [40]. Therefore, the killing of cancer cells through the ROS or the oxidative stress causing-agents represent one of the theories proposed in the cancer therapy.

Superoxide dismutase is the first defense barrier against ROS, which catalyzes the dismutation of superoxide anion radicals (O_2_^−^) to H_2_O_2_. Hydrogen peroxide generated by the activity of SOD is eliminated by its conversion into H_2_O in subsequent reactions by CAT and POD [41]. Similar studies have been found in the accessed literature, which prove the different actions of some known antioxidant polyphenols on the tumor cells, as compared to the normal ones.

Thus, Khan and colleagues (2013) evaluated the effects of resveratrol (RSV)—the natural polyphenol known for its antioxidant action—on the activities and expression levels of antioxidant enzymes in cancer cells [5]. In this study, the tumor cells PC-3 (prostate cancer), HepG2 (liver cancer), MCF-7 (breast cancer) and the normal kidney cells HEK293T were treated with a wide range of RSV concentrations (10–100 pM) for 24–72 h. The main cell parameters were determined by methods similar to those of our present study (the staining of the cells was done with trypan blue, the activities of antioxidant enzymes were spectrophotometric determined, and the percentage of apoptotic cells was determined by flow cytometry).

Applying a low concentration of RSV (25 µM) significantly increased the SOD activity in PC-3, HepG2, and MCF-7 cells, but not in the normal HEK293T cells. Catalase activity was increased in HepG2 cells, but glutathione peroxidase (GPX) activity was not altered during RSV treatment. Resveratrol-induced SOD2 expression was observed in cancer cells, although the expression of SOD1, CAT and GPX1 was not affected.

Apoptosis increased after the treatment with RSV was applied to cancer cells, in particular to PC-3 and HepG2 [5]. Thus, the results of the study showed that RSV exclusively inhibits the growth of cancer cells, which are similar results to those of the present study, in which usnic acid has been shown to inhibit the tumor cell development.

Similarly, in our study, the stimulation of activity of the antioxidant enzymes at 400 µg/mL in context with the maximum cytotoxic effect of UBE at the same concentration on CAL 27 cell line could be explained by the production of a large amount of ROS in the tumor cells, which was induced by the dry extract [40]. The increased levels of the antioxidant enzymes activities can be explained as a defense strategy against the oxidative stress induced by UBE.

Thus, the highest concentration of UBE induces a maximum level of stimulation of the CAT and POD activity. Finally, the antioxidant defense is outweighed by the oxidative stress induced by UBE and the CAL 27 cells death occurs.

## 4. Materials and Methods

### 4.1. Lichen Samples

The lichen samples were harvested from Calimani Mountains, Romania, in March. The determination of the investigated species was realized using standard methods. The lichen was cleaned, dried in an airy room below 25 °C and stored in the same conditions [24].

### 4.2. Preparation of Lichen Extracts

Three samples of the dry lichen were each extracted with different solvents (water, acetone, and 96% ethanol). The three resulting extractive solutions were filtered and then made up to 100 mL volumetric flask with the each solvent used in the extraction [24]. The dry extract of *Usnea barbata* (L.) F. H. Wigg. was prepared by continuous reflux in acetone on Soxhlet, at 70 °C, followed by evaporation of the solvent [25]; it was stored in the freezer, at a temperature below 20 °C, until its use. The determination of the usnic acid content was performed by the high-performance liquid chromatography methods: HPLC for ethanol, aqueous, and acetone liquid extracts [24], and UHPLC (PerkinElmer, Inc., Waltham, MA 02451, USA) for acetone dry extract, redissolved in acetone and in DMSO [25].

### 4.3. Determination of Usnic Acid Content in Different Extracts of Usnea barbata (L.) F. H. Wigg.

#### 4.3.1. HPLC Analysis of Usnic Acid in Various Liquid Extracts of *Usnea barbata* (L.) F. H. Wigg

For the identification and quantification of usnic acid in the ethanol, aqueous and acetone extracts of *Usnea barbata* (L.) F. H. Wigg, it was used an Agilent Technologies HPLC instrument Zorbax XDB with C18 column (150 mm/4.6 mm, 5 µm) [24]. The mobile phase was methanol: water: acetic acid (80:15:5, *v*/*v*/*v*), and the detection was made at 282 nm. The standard was usnic acid (Sigma-Aldrich, St. Louis, MO 63103, USA) dissolved in acetone 50 µg/mL and it was injected 6 times (every 20 µL) in the chromatographic system; the flow rate was 1.5 mL/min, the temperature was 25 °C and the analysis time was 6 min. Identification and quantitative determination of active compounds in assayed solutions were performed by comparing the chromatogram of standard with that of solution to be analyzed. Usnic acid presented a retention time of 4.463 ± 0.008053 min, with a correlation coefficient r^2^ = 0.9998 [24].

#### 4.3.2. UHPLC Analysis of Usnic Acid in UBE

Identification and determination of usnic acid content in UBE dissolved in acetone and in DMSO was performed by UHPLC method [25]. The dry extract was analyzed with a Perkin-Elmer UHPLC instrument Flexar FX 20 with C18 column (150 mm/4.6 mm, 5 µm), a Binary LC Pump, PDA plus detector (PerkinElmer, Inc., Waltham, MA 02451, USA), thermostat compartment for the column, degassing system and auto-sampler. The mobile phase was an isocratic system/methanol/water/glacial acetic acid (80:15:5), and the detection was made at 282 nm. The samples were as follows: UBE solubilized in acetone and in DMSO 0.2%, diluted 1 to 10, 1 to 20, and 1 to 50. The reference substance was usnic acid (Sigma-Aldrich, St. Louis, MO 63103, USA) in acetone and in DMSO 0.2% at concentrations of 10, 20, 50, 100, 200 µg/mL, and it was injected in the chromatographic system at injection volume of 20 µL. The flow rate was 1.5 mL/min, the temperature value was 25 °C and the analysis time was 6 min; the calibration curves were drawn for acetone (y = 5.19752 × 10^4^x − 1.30654 × 10^5^; r^2^ = 0.999808) and DMSO (y = 4.84629 × 10^4^x − 4.01679 × 10^4^; r^2^ = 0.999877) [25].

### 4.4. Evaluation of UBE Cytotoxic Activity by BSL Assay

Brine shrimp larvae were obtained by introducing the cysts of *Artemia salina* L. in saline solution of 35‰, for 24 h, under conditions of continuous illumination and aeration. After hatching Brine shrimp larvae in the larval stage I (instar I), they were separated and introduced into experimental vessels (pots with volume of 1 mL), in 2–3‰ saline solutions.

For these tests, a stock solution of UBE was prepared by solubilization of dry extract in DMSO 0.1%. Six different UBE concentrations (C1–C6 µg/mL) in DMSO 0.1% were tested: C1 = 266 µg/mL, C2 = 200 µg/mL, C3 = 100 µg/mL, C4 = 70 µg/mL, C5 = 45 µg/mL, and C6 = 30 μg/mL. *Artemia salina* L. larvae were not fed during the test period in order not to interfere with the tested extracts. The test is valid for 24–48 h, during which the larvae have embryonic energy reserves. The organisms exposed to different concentrations of UBE were evaluated periodically, following the movements of the antennae as well as the passage of the larvae in stages II and III. Mortality rate was recorded after 24 and 48 h of exposure, being the quantified parameter of the response to the various concentrations of UBE from the experimental recipients. For control, 3‰ saline solution and 0.1% DMSO in saline solution were used to evaluate the effect of solvents on *Artemia salina* L. larvae. For each different tested concentration of UBE in DMSO 0.1% were performed 4 repetitions. The results were analyzed using the Probit analysis method (StatPlus:mac, AnalystSoft Inc.—statistical analysis program for macOS^®^. Version v7. See https://www.analystsoft.com/en/, Walnut, Ca).

### 4.5. Effects of UBE Treatment on Morphological Characteristics of CAL 27 (ATCC^®^ CRL-2095™) Cell Line

Morphological changes of CAL 27 cells after treatment with different concentrations (12.5–400 µg/mL) of UBE were observed on a NIKON Eclipse TS100 inverted microscope. Images were acquired through the 10x objective and taken using an MS60–2 6.3MP sCMOS Camera.

### 4.6. In Vitro Cytotoxicity of UBE on CAL 27 (ATCC^®^ CRL-2095™) Cell Line

*Cell line, cell culture*. The CAL 27 cells were cultured in Dulbecco’s Modified Eagle’s Medium (DMEM) supplemented with 10% fetal bovine serum, penicillin (100 IU/mL) and streptomycin (100 µg/mL) in a humidified atmosphere of 5% CO_2_ at 37 °C, in a Binder incubator, until confluence. After 24 h, the cells were dissociated with trypsin-ethylenediamine tetraacetic acid (trypsin-EDTA), counted using a Cellometer Mini Automated Cell Counter (Nexcelom Bioscience), and the cell viability was assessed by the trypan blue dye exclusion method. Then, the cells were cultured in 96-well plates (TPP Techno Plastic Products AG, Trasadingen, Switzerland) with a density of 8 × 10^3^ cells/well, being incubated under the same temperature and humidity conditions in the binder incubator.

After 24 h, which is necessary for monolayer formation, the cells of the different experimental variants were treated, for 24 and 48 h respectively with UBE, which was dissolved in DMSO 0.2% for obtaining six final concentrations. The negative control was performed using growth medium alone, and the impact of the solvent was tested.

#### 4.6.1. MTT Assay

The MTT colorimetric method, modified after Mosmann, 1983 [42] and Laville et al., 2004 [43] involves as a biochemical mechanism, the NAD(P)H-dependent cellular oxidoreductase enzyme that converts the yellow tetrazolium [3-(4, 5-dimethylthiazolyl-2)- 2,5-diphenyltetrazolium bromide] (MTT) into insoluble (*E*,*Z*)-5-(4,5-dimethylthiazol-2-yl)- 1,3-diphenyl formazan (formazan) [44]. The formed formazan can be dissolved with DMSO to give a purple color with characteristic absorption. The intensity of the purple color is directly proportional to the living cell number, thus indicating the cell viability [45]. This quantitative, sensitive, and very precise method evaluates the effect of the tested solutions on the cell viability; it is suited for adherent cell cultures, allowing the processing of a large number of samples, the recorded absorption converting into cell numbers based on standard curves constructed with known cell dilutions [46]. The absorbance was evaluated using the Biochrom EZ Read 400 microplate automatic reader at 570 nm. The cell viability percentage was calculated according to the equation:Cell viability (%) = (Abs.) Test/(Abs.) Control × 100,
where Abs is the absorbance.

#### 4.6.2. Apoptosis Assay

For CAL 27 cells apoptosis evaluation, cells grown in 12-well plates and a density of 8 × 10^4^ cells/well were used. After the cell monolayer formation, the treatment with UBE was applied, in doses of 100 and 200 µg/mL, respectively, for 6 and 24 h. Subsequently, the cells were processed according to Annexin V-Fluorescein isothiocyanate/Propidium Iodide assay (Annexin V-FITC/PI assay) [47] that consist in a strong affinity of Annexin V for phosphatidylserine residues (normally hidden within the plasma membrane) on the surface of the cell [48]. Throughout the apoptosis process, the phosphatidylserine is translocated from the internal to the external cell surface. Propidium iodide was used to differentiate the dead cells from the living ones, and by association with Annexin V, it discriminated between the pre-apoptotic and apoptotic cells [49]. Summarily, the cells were detached by trypsinization, washed with cold phosphate-buffered saline (PBS), re-suspended in binding buffer, marked with Annexin V-FITC and propidium iodide (eBioscience kit) and, afterwards, the fluorescence was collected using specific filters for FITC and PI with a flowcytometer (Beckman Coulter Cell Lab QuantaSC). The data were analyzed with the FCSAlyzer v0.918-alpha [50].

#### 4.6.3. Assessment of Antioxidant Enzyme Activity

In order to reveal the substrate of cytotoxic effect of UBE on CAL 27 cell line, the activity of the main antioxidant enzymes, superoxide dismutase, catalase, and peroxidase was evaluated [51].

For antioxidant enzyme activity evaluation, CAL 27 cells were grown in Dulbecco’s Modified Eagle’s Medium supplemented with 10% fetal bovine serum, penicillin (100 IU/mL), and streptomycin (100 µg/mL) in a humidified atmosphere of 5% CO_2_ at 37 °C, in a binder incubator. After the monolayer formation, the cells were submitted to the oxidative stress by treatment for 15 min with 100 µM H_2_O_2_; after that, were twice washed with cold PBS solution and we applied the treatment with UBE in doses of 100 and 400 µg/mL, respectively, for 24 h. Subsequently, the cells were trypsinized and processed to obtain the enzymatic lysates used in the quantification of the antioxidant enzymes activity [52].

##### Determination of Superoxide Dismutase Activity

Superoxide dismutase activity was measured in accordance with Winterbourne’s assay with small modifications, and it is based on the ability of the enzyme to inhibit the reduction of nitro blue tetrazolium (NBT) by superoxide radicals generated through the reoxidation of photochemically reduced riboflavin. The degree of inhibition produced by the enzyme under standard conditions was estimated by determining the sample and control extinctions at 562 nm relative to distilled water [53].

##### Determination of Catalase Activity

Catalase activity was determined by Sinha’s assay, with minor adaptations [54]; CAT is allowed to act on oxygenated water for a fixed period of time, after which the enzyme is inactivated by the addition of a mixture of potassium dichromate–acetic acid. After stopping the action of catalase, the amount of unchanged oxygenated water reduces, in the acidic medium, the potassium dichromate to chromic acetate, which is determined at 570 nm. The difference between the initial and final amount of oxygenated water in the reaction medium represents the amount of oxygenated water decomposed by catalase.

##### Determination of Peroxidase Activity

Peroxidase activity was evaluated by spectrophotometric method based on measuring the color intensity of the product of ortho-dianisidine oxidation using oxygenated water, under the action of POD, at the wavelength of 562 nm [54].

The activity of each enzyme was expressed as enzyme units per g protein.

### 4.7. Statistical Analysis

All the in vitro experiments were performed based on three repetitions and statistically analyzed using the Student’s t test. The obtained values are expressed as mean ± SE of the three parallel measurements [55].

## 5. Conclusions

It was proven that the tested dry extract of *Usnea barbata* (L.) F. H. Wigg had an important cytotoxic effect on *Artemia salina* (L.) larvae, which was confirmed also by its strong in vitro cytotoxicity on human tongue squamous cells carcinoma—CAL 27 (ATCC^®^ CRL-2095™) cell line—through an apoptotic mechanism.

The results of this study could recommend this species of lichens from Romania as a source of natural phenolic compounds useful in anticancer therapy, alone or in association with other standard chemotherapeutics.

## Figures and Tables

**Figure 1 molecules-25-01865-f001:**
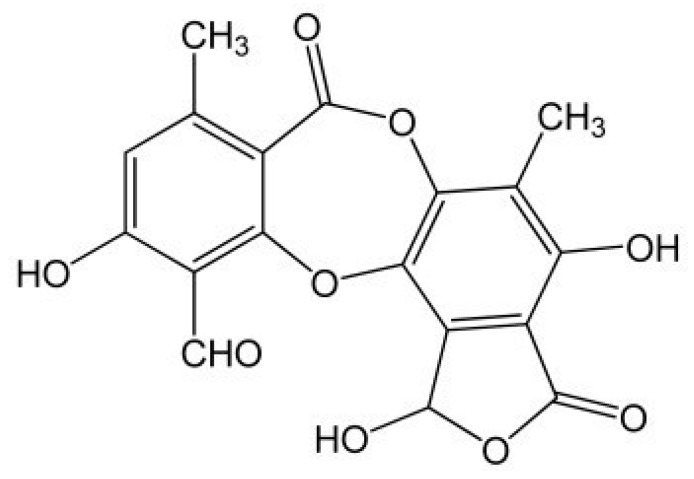
Chemical structure of usnic acid.

**Figure 2 molecules-25-01865-f002:**
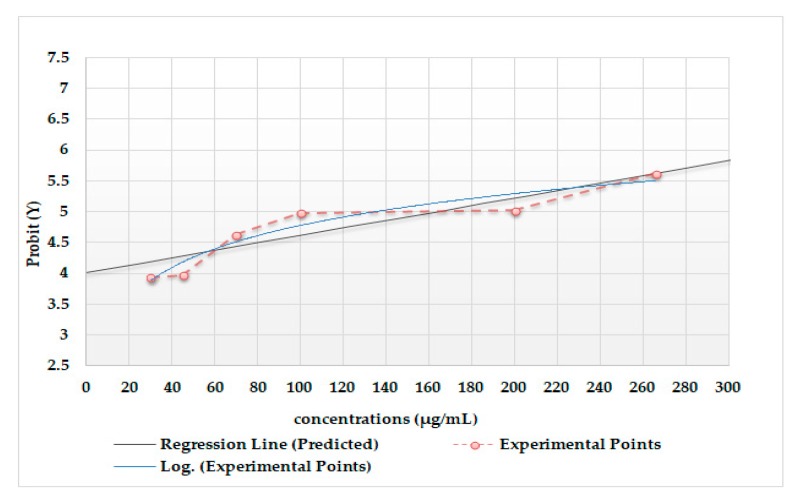
Concentration–effect correlation through Probit analysis for *Usnea barbata* (L.) F. H. Wigg. dry extract using Brine Shrimp Lethality assay (regression statistics and predicted points: LC_50_ = 164.92 µg/mL; LD_50_ Standard Error = 13.70 µg/mL; LD_50_ LCL = 129.40 µg/mL; LD_50_ UCL = 200.43 µg/mL; LC_80_ = 330.13 µg/mL; LC_100_ = 412.74 µg/mL).

**Figure 3 molecules-25-01865-f003:**
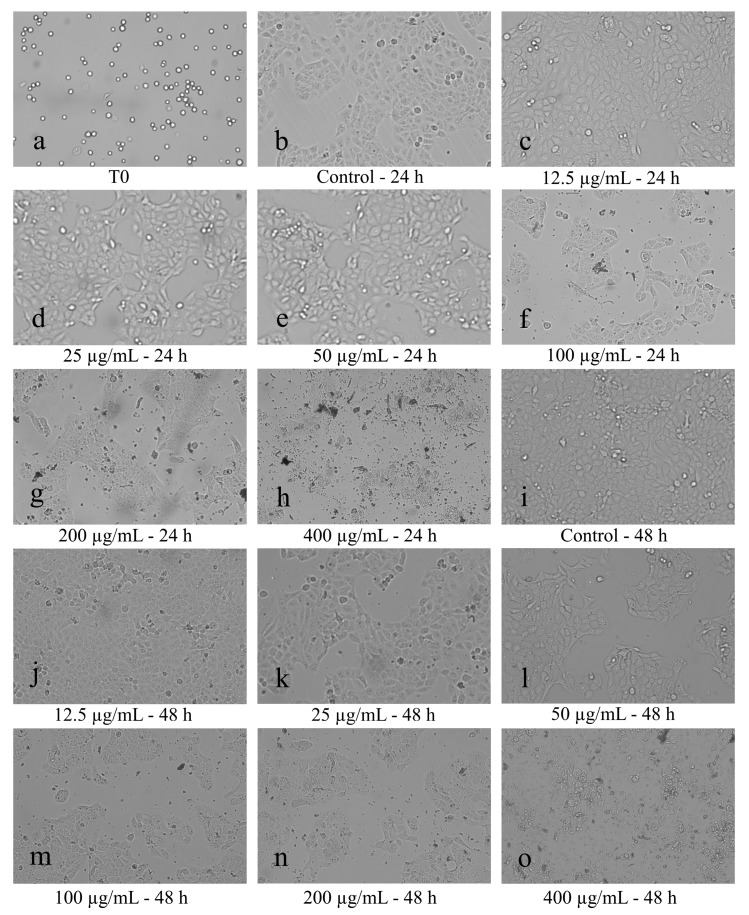
Cell morphology after UBE treatment on CAL 27 cells, after 24 h (**c**–**h**) and 48 h (**j**–**o**): **a**—Initial cultivation (T0); **b**,**i**—Control; **c**,**j**—12.25 µg/mL; **d**,**k**—25 µg/mL; **e**,**l**—50 µg/mL; **f**,**m**—100 µg/mL; **g**,**n**—200 µg/mL; **h**,**o**—400 µg/mL)

**Figure 4 molecules-25-01865-f004:**
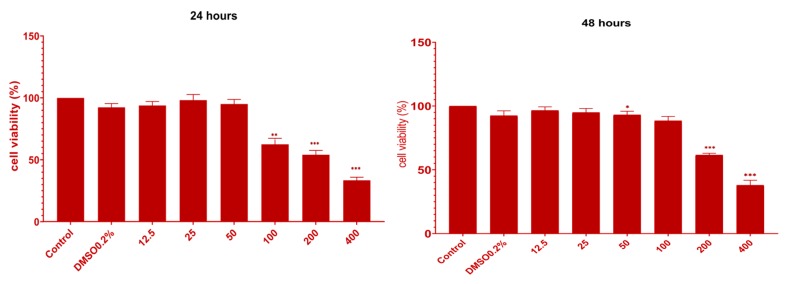
The viability of the CAL 27 cells after UBE (µg/mL) treatment for 24 h (left) and 48 h (right). The results represent the mean ± SE of three independent experiments (* *p* < 0.05, ** *p* < 0.01 and *** *p* < 0.001) when comparing the effects of UBE with the untreated control (t-test).

**Figure 5 molecules-25-01865-f005:**
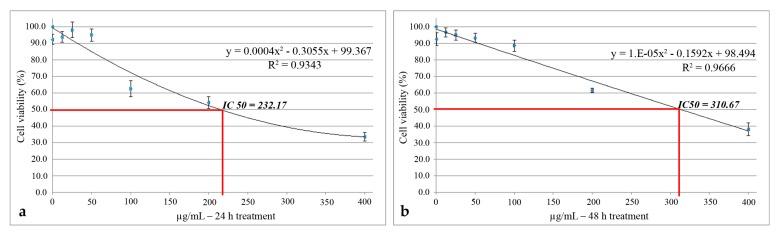
IC_50_ values after 24 h (**a**) and 48 h of (**b**) UBE treatment on CAL 27 cells.

**Figure 6 molecules-25-01865-f006:**
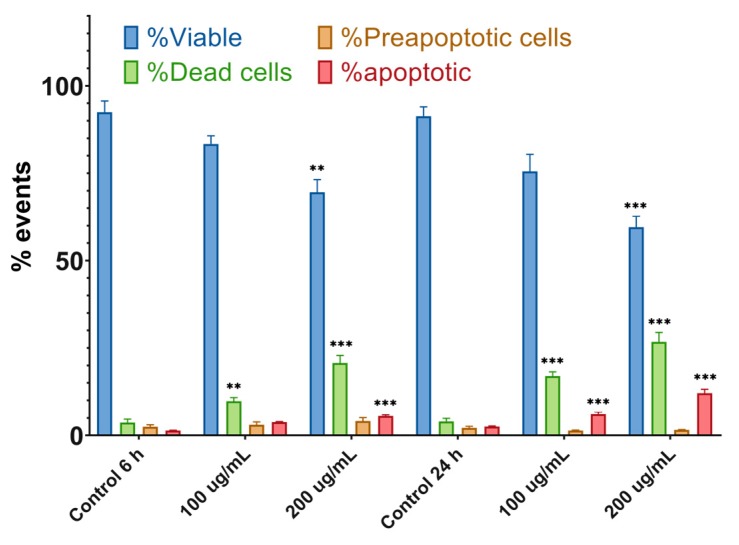
The frequency of distribution of viable, dead, pre-apoptotic and apoptotic cells in UBE-treated CAL 27 cell cultures, at concentrations of 100 and 200 µg/mL. These results represent the mean ± SE of three independent experiments (** *p* < 0.01 and *** *p* < 0.001) when comparing the effects of UBE with the untreated control (t-test).

**Figure 7 molecules-25-01865-f007:**
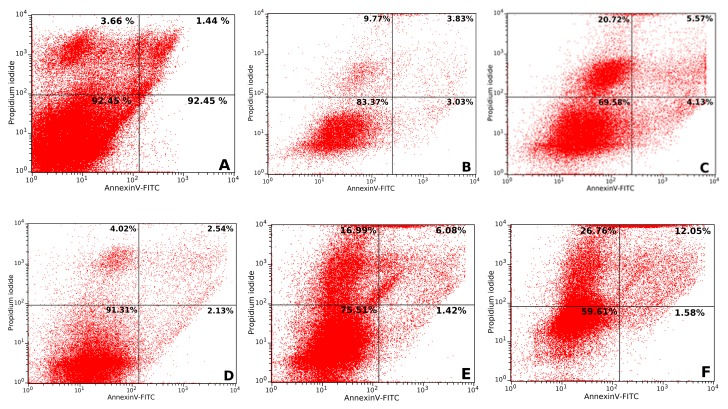
The flow cytometric cytograms corresponding to apoptosis assay: (**A**) control 6 h; (**B**) 100 µg/mL 6 h; (**C**) 200 µg/mL 6 h; (**D**) control 24 h; (**E**) 100 µg/mL 24 h; (**F**) 200 µg/mL 24 h. These data represent the frequency distribution of the viable (Q1—left down), death (Q2—left up), apoptotic (Q3—right up), and pre-apoptotic (Q4—right down) cells corresponding to each quadrant (Q).

**Figure 8 molecules-25-01865-f008:**
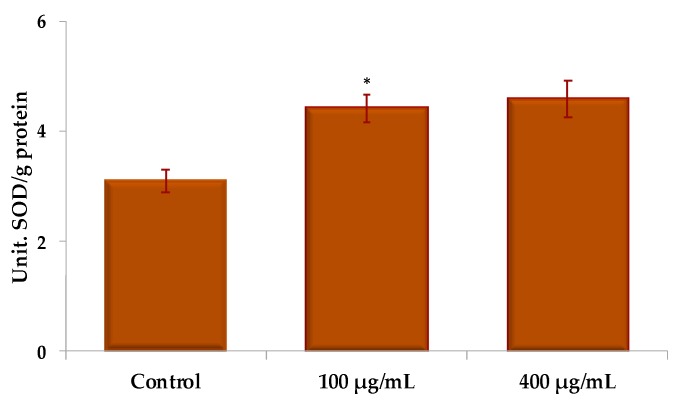
Evaluation of SOD activity (U SOD/g protein) under action of different UBE concentrations on CAL 27 cells. The results represent the mean ± SE of three independent experiments (* *p* < 0.05) when comparing the effects of UBE with the untreated control (t-test).

**Figure 9 molecules-25-01865-f009:**
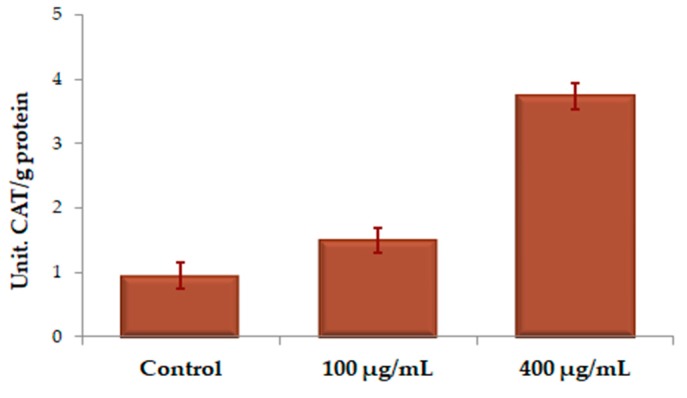
Assessment of CAT activity (U CAT/g protein) under action of different UBE concentrations on CAL 27 cells. The results represent the mean ± SE of the three independent experiments when comparing the effects of UBE with the untreated control (t-test).

**Figure 10 molecules-25-01865-f010:**
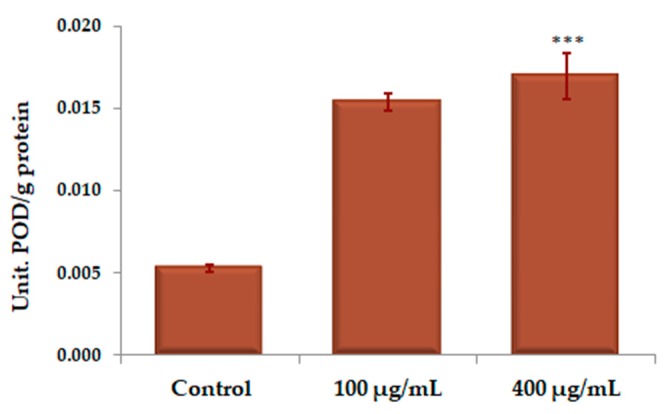
Determination of POD activity (U POD/g protein) under action of different UBE concentrations on CAL 27 cells. The results represent the mean ± SE of three independent experiments (*** *p* < 0.001) when comparing the effects of UBE with the untreated control (t-test).

**Table 1 molecules-25-01865-t001:** Usnic acid content (g %) in various extracts of *Usnea barbata* (L.) F. H. Wigg.

*Usnea barbata* (L.) Extract	Usnic Acid Content (g %)	Method
Aqueous extract	0.04	HPLC
Ethanol macerate	0.26	HPLC
Acetone macerate	2.12	HPLC
Dry acetone extract in DMSO	16.53	UHPLC
Dry acetone extract in acetone	31.59	UHPLC

**Table 2 molecules-25-01865-t002:** Comparative antioxidant enzymes activity corresponding to UBE treatment

Contact Solution	U SOD/g Protein	U CAT/g Protein	U POD/g Protein
Control	3.111 ± 0.511	0.944 ± 0.274	0.0053 ± 0.0002
UBE 100 µg/mL	4.429 ± 0.550	1.499 ± 0.287	0.0154 ± 0.0005
UBE 400 µg/mL	4.599 ± 0.627	3.735 ± 0.418	0.0170 ± 0.0014

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
