# Peer review of "Evaluation of the Cytotoxic Activity of the Usnea barbata (L.) F. H. Wigg Dry Extract"

_molecules, 2020, doi:10.3390/molecules25081865_

Round 1

Reviewer 1 Report

After the revision, the manuscript was improved. However, I still think only cytotoxicity effect on CAL 27 is not enough. A comparison effect on normal cell was needed and it could become more convincing.

Why the usnic acid content was more in acetone than that in DMSO? Why the change of solution could affect the content?

The layout of Tables should revise.

The paragraphs of discussion and conclusion were too many.

Author Response

Dear Reviewer 1,

First, we are glad to show our gratitude for the excellent communication  with the Academic Editor, our Assistant Editor, and our rewiewers. You are a great team and we appreciate that you have given us the opportunity to improve our manuscript. You can find our responses in the attached document.

Best regards,

Violeta Popovici

Reviewer 2 Report

I would like to thank the authors for addressing most of the Rewier's concerns.

However, there are few points that still need to be addresed:

1. Overall the statistical analysis used seem not to be appropriate. Specifically, student t-test cannot be used to compare more than two groups as for figure 3, 5, 7, 8, and 9. Please ammend.
2. In Figure 8, the statistical significance is not shown, while the authors specify it both in the main text and figure legend.
3. Figure 4, please specify wheter each dot in the IC50 curve is or not a mean of N experiments. If yes, please indicate the SD on the graph.

Author Response

Dear Reviewer 2,

Our team sincerely appreciate the excelent communication with the Academic editor, the Assistant Editor and the both Reviewers. Thank you very much for the opportunity to improve our manuscript.

Our responses to your comments are found in the attachement. We respectfully invite you to read this document; we are looking forward to your replay.

Kind regards,

Violeta Popovici

This manuscript is a resubmission of an earlier submission. The following is a list of the peer review reports and author responses from that submission.

Round 1

Reviewer 1 Report

The study by Violeta Papovici et al. describes the cytotoxic activity of Usnea barbata (L.) F. H. Wigg dry acetone extract on both BSLA and in vitro assay. The authors speculate that the extract induces cells death by promoting apoptosis and high oxidative stress, as they state in the abstract. However, in the main text, the authors show that the extract has antioxidant effect as it induces the expression of CAT, SOD and POD. This aspect must be be clarified.   

Moreover, there are several points that need to be addressed:

  1. In the introduction (lines 45-48), the authors describe the dual role of ROS in cancer progression. This concept should be better explained to let the reader to understand why they are looking for a natural product endowed with antioxidant properties.
  2. Results: the major concern is about the statistical analysis. The authors state several times that differences are or are not statistically significant without performing any statistical analysis.

Specifically:

  1. Fig 2A, 2B, Figures 6, 7, 8, and 9: p value and the type of analysis performed are missing;
  2. Fig 2A and B: to say that there are differences in terms of cell viability between 24 and 48 h, they should compare and analyze the two graphs. Moreover, is not clear to me why the cytotoxic activity at 48h are lower than 24h. They should measure and analyze cell viability up to 72h by both MTT assay and cell counting;
  3. Fig 3A and 3B: SD and the type of analysis performed are missing;
  4. Fig 4: p value, SD and the type of analysis performed are missing.
  5. Figure 5. The authors should specify that the dot plot is a representative plot of N experiments. I have many concerns about the appropriateness of FACS analysis, as the dot plot seems to be not properly compensated (see panel d). In each quadrants there are more than one population. This could mystify the results. Moreover, the authors should indicate the percentage of population in each quadrants.   
  6. The Bradford protein assay has been used to assess cytotoxicity. However, in my opinion, this method is not enough specific and sensitive to be used as surrogate of cell viability and should be deleted.
  7. The authors should unify the experiments in terms of ug/ml of extract used. In Fig 4 they used 100 and 200 ug/ml, while in Fig 6 to Fig 9 they used 100 and 400 ug/ml.
  8. The authors should analyze whether a correlation between the antiapoptotic and antioxidant effect exists to understand if these are both responsible of cell death induced by the extract.
  9. Overall, the quality of the figures is poor (for example the title for x- and y-axis is missing in Fig 4, and Fig 6 to Fig 9) and the figure description in the legend is lacking.    

Reviewer 2 Report

In this paper, the authors tested the usnic acid content of extracts of Usnea barbata (L.) F. H. Wigg. then, the cytotoxicity of UBE was evaluated using CAL 27 cell line with MTT assay and Apoptosis assay. Protein assay and antioxidant enzyme activities were also tested. I think the experimental design is too simple. Many related points were not discussed clearly. The reported results are routine and preliminary. Some detail comments are listed as follow.

  • Many of figures are not clear enough. It looks like they were simply captured instead of output.
  • The chemical analysis is too simple. Only usnic acid was tested. HPLC and UHPLC could give much more information than that.
  • Since the experimental section is the last part, some abbreviation of words should be mentioned at the first place in paper.
  • In section 2.3.1, after 48 h the cell viability slightly increased compared with that after 24 h. I think the reason given by author is unconvincing. Maybe an investigation for the effect of incubation time on the cell viability could give us more clues.
  • In section 2.3.2, the investigated time are 6 and 24 h, not 24 h and 48 h. Two related investigations (cell viability and Apoptosis) using different time range. I think it reduced the values and comparability of data.
  • I think author should make the experiments on a normal cell line as a contrast. From the available result, I could only know the extract could induce the CAL 27 cells death and high oxidative stress. It’s useless if this extract has the same effects on normal cell.